# Cross-cultural adaptation and validation of The Resilience Scale for Kidney Transplantation (RS-KTPL) in a Chinese population

Yan Tang[1,2]*, Jiefu Zhu[1,2], Tao Qiu[1,2], Shan Qiu[3], Wei Lei[3], Xiaolan Mao[3]*

1 The Department of Organ Transplantation, Renmin Hospital of Wuhan University, Wuhan, Hubei, China, 2 The Department of Urology, Renmin Hospital of Wuhan University, Wuhan, Hubei, China, 3 The Nursing Department, The Central Hospital of Enshi Tujia and Miao Autonomous Prefecture, Enshi, Hubei, China

* 1739883176@qq.com (XM); 1348580254@qq.com (YT)

## Abstract

### Objective

This study aimed to translate and validate The Resilience Scale for Kidney Transplantation (RS-KTPL) into Chinese and assess its reliability and validity among kidney transplant patients in China.

### Methods

With authorization from the original authors, the RS-KTPL was translated following Brislin's translation model, including forward translation, back translation, author review, cross-cultural adaptation, and a pilot study, resulting in a Chinese version of the RS-KTPL. A total of 358 kidney transplant recipients were recruited through convenience sampling and completed the questionnaire. Statistical analyses included item analysis, content validity, structural validity, convergent validity, discriminant validity, and reliability.

### Results

Item analysis led to the removal of certain items that did not meet the criteria, resulting in a final version of the scale with 22 items across four dimensions. For content validity, the item-level content validity index (I-CVI) ranged from 0.83 to 1.000, and the scale-level content validity index (S-CVI/Ave) was 0.91, indicating good content fit. Structural validity was confirmed through exploratory and confirmatory factor analyses, supporting a four-factor structure with a cumulative variance contribution rate of 64.913% and all factor loadings exceeding 0.5. Convergent and discriminant validity analyses showed that the composite reliability (CR) values ranged from 0.741 to 0.938, and the average variance extracted (AVE) values ranged from 0.5 to 0.704, with the square root of AVE being higher than the inter-factor correlation coefficients,

**Data availability statement:** All relevant data are within the manuscript and its Supporting Information files.

**Funding:** This work was supported by grants from the National Natural Science Foundation of China (82100803) and Knowledge Innovation Program of Wuhan-Shuguang Project (2023020201020505).

**Competing interests:** The authors have declared that no competing interests exist.

indicating good internal consistency and discriminating ability. Reliability testing showed a Cronbach's α coefficient of 0.944 for the overall scale, with subscale Cronbach's α coefficients all above 0.696, and a split-half reliability of 0.891, demonstrating high internal consistency and stability of the scale.

## Conclusion

The Chinese version of the RS-KTPL exhibits good reliability and validity among kidney transplant patients in China and can be effectively used to assess psychological resilience in this population.

---

## Introduction

Kidney transplantation is an effective treatment for end-stage renal disease, significantly improving patients' quality of life and prognosis. However, recipients face multifaceted postoperative challenges, including surgical complications, immunosuppressant side effects, and psychological stress, which may lead to mental health issues and hinder recovery [1,2]. Psychological resilience—defined as the ability to adaptively cope with and recover from adversity [3]—plays a critical role in mitigating these challenges. Studies indicate that resilient patients exhibit better treatment adherence and self-management behaviors, ultimately enhancing clinical outcomes [4].

While resilience scales such as the Connor-Davidson Resilience Scale (CD-RISC) [5] and the Resilience Scale (RS) [6] are widely used in general populations, they lack specificity to the physiological and psychological needs of kidney transplant recipients. Moreover, their psychometric properties remain unvalidated in this population, highlighting the need for a tailored assessment tool.

The Resilience Scale for Kidney Transplantation (RS-KTPL)(2), developed in Korea in 2024, is the first scale designed specifically for this population. It addresses clinical gaps by integrating domains such as positive thinking, medical management, and healthcare relationships [2,7]. However, its applicability to Chinese patients remains unexplored. This study aimed to cross-culturally adapt and validate the Chinese RS-KTPL, providing a reliable tool for assessing resilience in this context.

## Materials and methods

### Scale translation

**Scale overview.** The Resilience Scale for Kidney Transplantation (RS-KTPL) was developed by Korean scholars Mi Ha Chung and Hyojung Park in 2024 [2]. The original scale consists of 27 items across six dimensions: Positive Thinking and Adaptation (11 items), Supportive Interpersonal Relationships (4 items), Self-awareness of Negative Psychological Responses (3 items), Physical Health Control (4 items), Homeostasis Regulation (2 items), and Supportive Relationships with Healthcare Professionals (3 items). The scale uses a 4-point Likert scoring system, ranging from 1 (Strongly Disagree) to 4 (Strongly Agree). The total score is the sum

of all item scores, with higher scores indicating greater psychological resilience. The overall Cronbach's α coefficient of the original scale was 0.87, with subscale Cronbach's α values ranging from 0.57–0.86.

**Scale translation process.** In November 2024, the researchers contacted the original authors via email and obtained permission to translate the RS-KTPL into Chinese. The translation followed Brislin's translation model and consisted of the following steps:

① Forward Translation: Two nursing researchers fluent in Chinese and holding master's degrees independently translated the original scale into two Chinese versions (S1 and S2). The research team discussed the discrepancies and reached a consensus, forming the initial Chinese draft (S3).

② Back Translation: Two university English professors without medical backgrounds independently back-translated the Chinese version into English (H1 and H2). Items with an agreement rate below 90% were revised and retranslated until a consensus was reached, forming the final back-translated version (H3).

③ Author Review: The back-translated version (H3) was sent to the original authors for review. The research team discussed and incorporated their feedback, producing a refined forward-translated version (S4).

**Cross-cultural adaptation.** A cross-cultural adaptation panel consisting of 12 experts from Wuhan University School of Nursing, Renmin Hospital of Wuhan University, and Enshi Central Hospital was formed. The panel included two experts in transplant nursing, two experts in psychiatric nursing, two experts in nursing management, one expert in rehabilitation nursing, one expert in chronic disease care, three transplant physicians, and one specialist in psychiatric medicine. The 12 experts had a mean age of $47.2 \pm 8.8$ years (range: 35–59 years) and $23.67 \pm 13.34$ years of experience in their respective fields. Their academic qualifications included 4 with bachelor's degrees (33.3%), 3 with master's degrees (25%), and 5 with doctoral degrees (41.67%). The panel independently reviewed the translated items and provided suggestions for their cultural adaptation. Revisions were made accordingly, and the experts were consulted again until a culturally adapted Chinese version (S5) of the RS-KTPL was finalized.

## Participants and study design

**Pilot study.** In January 2025, 20 post-kidney transplant inpatients at Renmin Hospital of Wuhan University were recruited for a pilot study. Patients were asked to complete the Chinese RS-KTPL and provide feedback on item comprehension, response formats, and questionnaire instructions. The completion time ranged from 5 to 8 min. All participants reported that the items were clear and easy to understand, and no modifications were required.

**Formal investigation.** A convenience sampling method was used to recruit kidney transplant recipients from the Renmin Hospital of Wuhan University as research participants. The inclusion criteria were: (1) recipients aged >18 years, (2) post-transplant ≥3 months with stable renal function, and (3) normal reading, writing, and communication abilities; (4) capable of independently or with guidance completing the questionnaire; and (5) voluntary participation with informed consent. The exclusion criteria were as follows: multiple organ transplantation, pregnancy or lactation, major surgery within the past month, and cognitive impairment. This study complied with the Declaration of Helsinki, and the research protocol was approved by the Ethics Committee of Renmin Hospital of Wuhan University (Approval No. WDRY2024-K282).

## Research methods

**Data collection method.** A questionnaire survey was conducted. Before the survey, the researchers explained the study's purpose, content, and questionnaire requirements to the participants. Standardized instructions were used to clarify questions when necessary to ensure response accuracy. After obtaining informed consent, the survey was conducted anonymously via Questionnaire Star, where participants scanned a QR code with an electronic device

and completed the questionnaires. If the participants had difficulty understanding the items, the researcher provided explanations without using suggestive or biased language. Responses were excluded if the completion time was less than 3 min or if all items were answered identically, indicating careless responses.

**Sample size calculation.** Following cross-cultural adaptation guidelines, the sample size should be at least ten times the number of items in the scale. The scale consists of 27 items across six dimensions, and considering 10%–20% invalid responses, the minimum required sample size (Nmin) was calculated as: $(27 \times 10) \times (1 + 10\%) = 297$; The maximum required sample size (Nmax) was:

$$(27 \times 10) \times (1 + 20\%) = 324$$

According to methodological guidelines, a minimum of 150 samples is required for exploratory factor analysis (EFA), while confirmatory factor analysis (CFA) necessitates at least 200 samples. To ensure robust analysis, we targeted a sample size of 350 [8,9]. In this study, 358 valid questionnaires were collected, yielding a response rate of 100%.

## Statistical analysis

The data were organized in Excel, reviewed by two researchers for logical errors, and analyzed using SPSS 26.0 and AMOS 24.0. Continuous variables are expressed as mean ± standard deviation ($\bar{x} \pm s$). Categorical variables were described using frequency and percentage. The psychometric properties of the scale were evaluated as follows:

**Item analysis.**

① Critical Ratio (CR) Method: The total questionnaire scores were ranked in a descending order. The top 27% (high-score group) and bottom 27% (low-score group) were compared using independent sample t-tests. Items with CR (t) <3 or non-significant differences were removed [8].

② Normality Assessment: Prior to correlation analysis, normality assumptions were verified through skewness/kurtosis statistics (all absolute values ≤2) and Kolmogorov-Smirnov tests (all p > 0.05), confirming suitability for parametric tests.

③ Correlation Coefficient Method: The Pearson correlation coefficient between each item score and the total scale score was calculated. Items with r < 0.4 or P > 0.05 were deleted [10].

**Validity testing.**

① Content Validity: Twelve experts assessed whether the measurement tool matched the intended constructs. Content validity was evaluated using the Item-level Content Validity Index (I-CVI) and Scale-level Content Validity Index (S-CVI/Ave). A scale was considered to have good content validity if I-CVI ≥ 0.78 and S-CVI/Ave ≥ 0.90 [11].

② Structural Validity: Exploratory factor analysis (EFA) was conducted on a random 150-subject sample. If KMO > 0.8 and Bartlett's test of sphericity was significant (P < 0.05), EFA was deemed appropriate [12]. Principal component analysis with maximum variance orthogonal rotation was used, as preliminary analyses indicated low inter-factor correlations (r < 0.30), supporting the assumption of factor independence. This approach aligns with the original scale's methodology [2] and enhances clinical interpretability by minimizing cross-loadings. items with factor loadings of <0.5 were deleted.

Confirmatory factor analysis (CFA): Conducted on the remaining 208-subject sample, using maximum likelihood estimation to fit the CFA model. The fit indices and recommended thresholds were: Chi-square/df (CMIN/DF) <3; Root Mean Square Error of Approximation (RMSEA) < 0.08; Root Mean Square Residual (RMR) <0.08; Parsimonious Goodness-of-Fit Index (PGFI) > 0.5; Comparative Fit Index (CFI) > 0.9; Tucker-Lewis Index (TLI) > 0.9; Incremental Fit Index (IFI) > 0.9 [13]. These indices collectively evaluate the model fit and ensure its suitability.

 

③ Convergent validity: This assesses how well the observed indicators measure their intended constructs. Evaluated using: Factor Loadings (> 0.4 recommended); Composite Reliability (CR > 0.7 recommended); Average Variance Extracted (AVE > 0.5 recommended) [14].

④ Discriminant Validity: Determined whether observed indicators correlated more strongly with their respective constructs than with others. This was assessed by comparing the square root of the AVE for each construct with the inter-factor correlation coefficients, where the square root of the AVE should be greater than the inter-factor correlations [14,15].

**Reliability testing.** Reliability was assessed using internal consistency and split-half reliability.

① Internal Consistency: The Cronbach's α coefficient was calculated for each dimension and the overall scale. A value of α ≥ 0.7 was considered acceptable [16].

② Split-Half Reliability: Items were divided into two groups based on odd and even numbers, and a correlation analysis was performed. An acceptable split-half reliability was set at ≥ 0.7 [16].

## Results

### Cultural adaptation results

Based on feedback from 12 experts, the following modifications were made to the translated items: Item 16:"I have established stable bonds through interactions with others" was revised to "I have established stable relationships through interactions with others" to avoid semantic ambiguity in Chinese; Item 19: " I always adhere to dietary therapy" was changed to "I always adhere to a healthy diet" to better reflect practical application; Item 25: "The compassion and comfort from medical staff give me strength" was modified to "The encouragement and comfort from medical staff give me strength" for a more accurate emotional expression; Additionally, Item 2 ("I have the confidence to overcome difficulties.") and Item 11 ("I am confident in overcoming any difficulties I encounter. "). were identified as being redundant. After discussion within the research team, Item 11 was removed, resulting in a final Chinese version of the RS-KTPL with 26 items.

### General patient characteristics

A total of 358 valid responses were collected. The demographic and clinical characteristics of the participants, including sex, age, ethnicity, employment status, place of residence, marital status, educational level, economic burden, and caregiving methods, are summarized in Table 1.

### Item analysis of the Chinese version of RS-KTPL

Correlation Coefficients (r): The initial correlation coefficients ranged from 0.116 to 0.761. After removing items 15, 16, and 17, the correlation coefficients improved to 0.371–0.761, all of which were statistically significant (P < 0.05). Critical Ratio Method: The top 27% of the total scores (115 participants) were classified as the high-score group, while the bottom 27% (95 participants) were classified as the low-score group. Independent sample t-tests showed that the critical ratio (CR) values for each item ranged from 0.362 to −22.24. After removing items 15, 16, and 17, the CR values ranged from 5.144 to −22.24, all reaching statistical significance (P < 0.05). Rationale for item removal: Items 15–17 (transplant-related anxiety) were excluded due to cultural differences in anxiety expression—Chinese patients' concerns are mitigated by high transplant success rates (92% 5-year survival) and often manifest somatically. Item 19 (fluid control) was removed because Chinese hospitals provide standardized hydration protocols, reducing the need for patient self-regulation (unlike the original Korean version). Thus, the final Chinese version of the RS-KTPL retained 23 items. The detailed results are presented in Table 2.

**Table 1. Complete demographic and clinical characteristics of the study cohort (N = 358).**

| Characteristic | Category | n | %/M ± SD |
|---|---|---|---|
| **Sex** | | | |
| | Male | 223 | 62.3 |
| | Female | 135 | 37.7 |
| **Age (years)** | | | 46.05 ± 10.33 |
| **Ethnicity** | | | |
| | Han | 339 | 94.7 |
| | Minority | 19 | 5.3 |
| **Employment Status** | | | |
| | Employed | 135 | 37.7 |
| | Retired | 70 | 19.6 |
| | Other | 153 | 42.7 |
| **Place of Residence** | | | |
| | Urban | 278 | 77.7 |
| | Rural | 80 | 22.3 |
| **Marital Status** | | | |
| | Single | 44 | 12.3 |
| | Married | 286 | 79.9 |
| | Other (Separated, Widowed, Divorced) | 28 | 7.8 |
| **Living Situation** | | | |
| | Living alone | 59 | 16.5 |
| | Living with others | 299 | 83.5 |
| **Education Level** | | | |
| | Primary school or below | 14 | 3.9 |
| | Middle school | 98 | 27.4 |
| | High school or vocational school | 89 | 24.9 |
| | College or undergraduate | 152 | 42.5 |
| | Graduate or above | 5 | 1.4 |
| **Monthly Household Income (CNY)** | | | |
| | <2000 | 74 | 20.7 |
| | 2000-5000 | 109 | 30.4 |
| | 5001-10000 | 97 | 27.1 |
| | >10000 | 78 | 21.8 |
| **Medical Insurance Type** | | | |
| | Self-paid | 4 | 1.1 |
| | Resident Medical Insurance | 73 | 20.4 |
| | Employee Medical Insurance | 222 | 62.0 |
| | Low-income subsidy | 50 | 14.0 |
| | Government-funded insurance | 2 | 0.6 |
| | Other | 7 | 2.0 |
| **Primary Caregiving Method** | | | |
| | Self-care | 233 | 65.1 |
| | Spouse care | 104 | 29.1 |
| | Child care | 3 | 0.8 |
| | Caregiver | 1 | 0.3 |
| | Other | 17 | 4.7 |

*(Continued)*

**Table 1.** (Continued)

| Characteristic | Category | n | %/M±SD |
|---|---|---|---|
| **Family Support Level** | | | |
| | None | 33 | 9.2 |
| | Low | 72 | 20.1 |
| | Moderate | 67 | 18.7 |
| | High | 186 | 52.0 |
| **Type of Kidney Transplant** | | | |
| | Deceased donor | 323 | 90.2 |
| | Living-related donor | 35 | 9.8 |
| **Post-Transplant Duration** | | | |
| | 3-<6 months | 30 | 8.4 |
| | 6-<12 months | 33 | 9.2 |
| | 1-<3 years | 107 | 29.9 |
| | 3-<5 years | 57 | 15.9 |
| | >5 years | 131 | 36.6 |
| **Repeat Kidney Transplantation** | | | |
| | Yes | 87 | 24.3 |
| | No | 271 | 75.7 |
| **Post-Transplant Complications** | | | |
| | Yes | 91 | 25.4 |
| | No | 267 | 74.6 |

Note. M = Mean; SD = Standard Deviation. All original variables and categories are presented without omission or modification. Percentages are calculated based on valid cases.

## Validity analysis of the scale

**Content validity.** Based on expert review (n = 12), the Item-Level Content Validity Index (I-CVI) ranged from 0.83 to 1.000, and the Scale-Level Content Validity Index (S-CVI/Ave) was 0.91, exceeding the 0.80 threshold. This indicates that the scale has good content validity and effectively reflects psychological resilience in patients undergoing kidney transplantation.

**Structural validity.** Exploratory factor analysis (EFA): The first 150 questionnaires were used for EFA. The KMO value was 0.916, and Bartlett's test of sphericity yielded a chi-square value of 2216.706 (P < 0.001), confirming that factor analysis was appropriate. As shown in Fig 1, four factors were extracted (cumulative variance contribution rate: 64.913%), supported by the scree plot's elbow criterion at eigenvalues >1.0. Factor loadings ranged from 0.508 to 0.810 across four identified domains: Positive Mindset (12 items), Medical Management (5 items), Social Support (3 items), and Healthy Habits (3 items). Visual support for factor extraction is provided in Fig 1 (Scree plot), which clearly demonstrates the elbow criterion at 4 factors with eigenvalues >1.0

As shown in Table 3, all items exhibited acceptable standardized loadings (> 0.5) on their respective factors without significant cross-loadings (all cross-factor loadings < 0.4), indicating a clear simple structure for the four-factor solution.

**Model fit testing.** Confirmatory factor analysis (CFA): The remaining 208 questionnaires were used for CFA. A four-factor model was tested using 23 observed variables as indicators. Initial model fit indices: CMIN/DF = 2.201, RMSEA = 0.076, RMR = 0.018, and CFI = 0.909.

To enhance model specification, modification indices were consulted, revealing that allowing error terms e14 and e15 to covary would significantly improve fit. This adjustment was theoretically justifiable as both items shared similar

**Table 2. Item analysis of the Chinese version of The Resilience Scale for Kidney Transplant patients (Mean±SD).**

| Item | High-Score Group (M±SD) | Low-Score Group (M±SD) | Critical Ratio (t) | Item-Total Correlation (r) |
|---|---|---|---|---|
| RS1 | 2.75±0.515 | 2.01±0.418 | 11.81** | 0.547** |
| RS2 | 2.90±0.295 | 2.00±0.316 | 22.24** | 0.743** |
| RS3 | 2.71±0.494 | 1.78±0.508 | 14.01** | 0.670** |
| RS4 | 2.78±0.416 | 1.86±0.394 | 17.03** | 0.726** |
| RS5 | 2.84±0.370 | 1.82±0.483 | 17.61** | 0.745** |
| RS6 | 2.78±0.480 | 1.87±0.407 | 15.30** | 0.733** |
| RS7 | 2.77±0.422 | 1.97±0.314 | 16.05** | 0.759** |
| RS8 | 2.71±0.494 | 1.96±0.351 | 13.06** | 0.681** |
| RS9 | 2.77±0.422 | 1.83±0.441 | 16.39** | 0.749** |
| RS10 | 2.87±0.342 | 1.95±0.425 | 17.68** | 0.714** |
| RS11 | 2.86±0.426 | 2.02±0.273 | 17.20** | 0.688** |
| RS12 | 2.87±0.342 | 1.86±0.394 | 20.38** | 0.761** |
| RS13 | 2.63±0.593 | 1.82±0.447 | 11.46** | 0.656** |
| RS14 | 2.69±0.487 | 1.88±0.369 | 13.78** | 0.710** |
| RS15 | 1.58±1.054 | 1.54±0.696 | 0.36 | 0.079 |
| RS16 | 1.69±1.050 | 1.61±0.711 | 0.61 | 0.116* |
| RS17 | 1.38±0.944 | 1.31±0.592 | 0.63 | 0.104* |
| RS18 | 2.65±0.519 | 1.87±0.464 | 11.83** | 0.610** |
| RS19 | 2.12±0.817 | 1.64±0.530 | 5.14** | 0.371** |
| RS20 | 2.45±0.620 | 1.90±0.374 | 7.88** | 0.494** |
| RS21 | 2.37±0.639 | 1.82±0.500 | 7.17** | 0.457** |
| RS22 | 2.85±0.411 | 2.04±0.374 | 15.33** | 0.646** |
| RS23 | 2.90±0.308 | 2.12±0.321 | 18.55** | 0.666** |
| RS24 | 2.77±0.486 | 2.02±0.365 | 13.05** | 0.626** |
| RS25 | 2.80±0.402 | 1.99±0.329 | 16.39** | 0.700** |
| RS26 | 2.91±0.281 | 2.09±0.408 | 17.39** | 0.649** |
| Total Score | 67.40±5.18 | 48.57±2.88 | 33.08 | 1.000 |

*Note. \*\*p<0.01, \*p<0.05. M=Mean; SD=Standard Deviation. High-score group (top 27%) n=115; Low-score group (bottom 27%) n=95. Items RS15 to RS17 were subsequently removed due to failing selection criteria (CR<3.0 and r<0.40).*

measurement contexts. The modified model demonstrated improved fit indices: CMIN/DF=2.054, RMSEA=0.071, RMR=0.016, CFI=0.920. all of which now met established thresholds for good model fit.

The final model's robustness was further evidenced by standardized loadings and factor relationships, as detailed in Fig 2 (CFA path diagram). Complete model fit statistics are systematically presented in Table 4.

**Composite reliability and convergent validity of the scale.** Based on the satisfactory model fit of the CFA model for the Psychological Resilience Scale, the composite reliability (CR) and convergent validity (AVE) of each dimension were analyzed. As Item RS19 had a standardized factor loading of 0.275, which did not meet the required threshold, it was removed from the scale.

As shown in Table 5, the CR values ranged from 0.741 to 0.938, and the AVE values ranged from 0.5 to 0.704, indicating that the scale had good composite reliability and convergent validity. The final Chinese version of the RS-KTPL included 22 items across four dimensions.

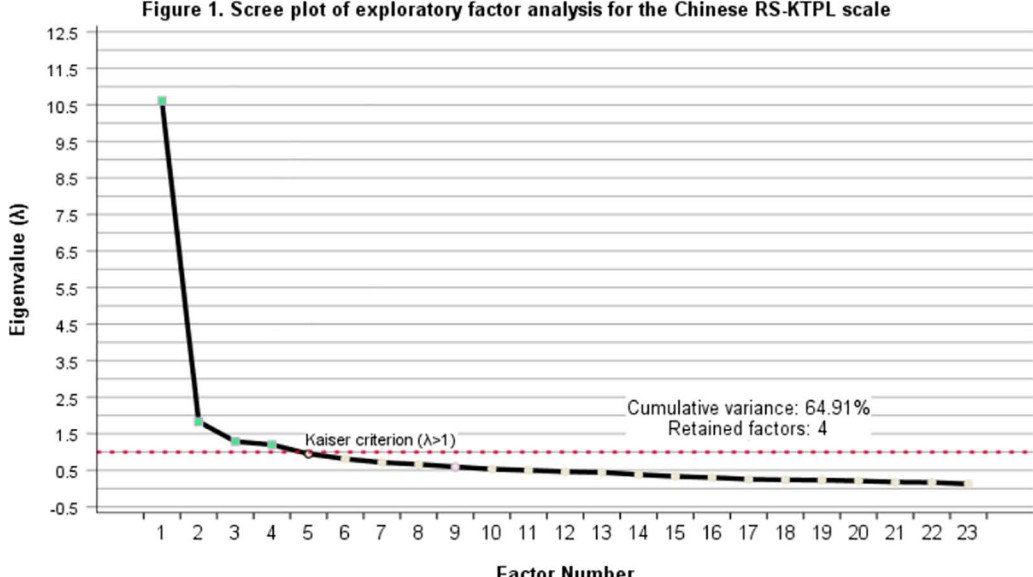

**Fig 1. Scree plot of exploratory factor analysis for the Chinese RS-KTPL scale.**

**Discriminant validity of the scale.** Discriminant validity assesses whether the observed indicators are more strongly correlated with their respective dimensions than with other dimensions. Table 6 shows that the square root of the AVE for each dimension is greater than the correlation coefficients among the different dimensions, confirming that the scale has good discriminant validity.

### Reliability analysis of the scale

Internal Consistency Reliability: Cronbach's α coefficient for the overall scale was 0.944. The Cronbach's α coefficients for the four dimensions were 0.939, 0.875, 0.801, and 0.696, respectively, indicating good reliability. Split-Half Reliability: The split-half reliability coefficient was 0.891. Conclusion: The scale demonstrated high reliability across all aspects.

## Discussion

### Reliability and validity of the Chinese version of the RS-KTPL scale and comparative analysis

The RS-KTPL scale is a psychological resilience scale specifically designed for kidney transplant recipients. Through the adaptation and psychometric validation of the Chinese version of this scale, healthcare professionals can more effectively identify patients who may face postoperative psychological challenges. Early identification enables physicians to develop personalized psychological support plans to enhance patients' postoperative adaptation and quality of life [17]. The findings of this study indicate that the adapted scale meets acceptable standards in terms of structural, convergent, and discriminant validity and reliability. Regarding structural validity, exploratory factor analysis (EFA) and confirmatory factor analysis (CFA) confirmed the four-dimensional structure of the scale, including positive mindset, medical management, social support, and healthy habits, with a cumulative variance contribution rate of 64.913%. This suggests that the scale adequately reflects the psychological resilience structure of patients undergoing kidney transplantation.

The analysis of convergent and discriminant validity showed that the composite reliability (CR) values for all dimensions exceeded 0.7, the average variance extracted (AVE) values were all above 0.5, and the square roots of the AVE

**Table 3. Standardized factor loadings of the Chinese RS-KTPL scale items.**

| Item | Positive Mindset | Medical Management | Social Support | Healthy Habits |
|------|------------------|--------------------|----------------|----------------|
| RS5 | 0.804 | – | – | – |
| RS3 | 0.779 | – | – | – |
| RS6 | 0.757 | – | – | – |
| RS12 | 0.756 | – | – | – |
| RS7 | 0.742 | – | – | – |
| RS2 | 0.734 | – | – | – |
| RS9 | 0.730 | – | – | – |
| RS8 | 0.728 | – | – | – |
| RS4 | 0.714 | – | – | – |
| RS10 | 0.679 | – | – | – |
| RS1 | 0.670 | – | – | – |
| RS11 | 0.588 | – | – | – |
| RS26 | – | 0.810 | – | – |
| RS23 | – | 0.799 | – | – |
| RS22 | – | 0.782 | – | – |
| RS25 | – | 0.768 | – | – |
| RS24 | – | 0.716 | – | – |
| RS13 | – | – | 0.762 | – |
| RS19 | – | – | 0.685 | – |
| RS14 | – | – | 0.658 | – |
| RS20 | – | – | – | 0.754 |
| RS21 | – | – | – | 0.720 |
| RS18 | – | – | – | 0.508 |

*Note. All factor loadings were statistically significant ($p < 0.001$). Dash (-) indicates items not loading on the respective factor. Loadings >0.50 are considered acceptable.*

were greater than the corresponding correlation coefficients, further confirming the scale's strong internal consistency and discriminant validity. The reliability test results demonstrated a Cronbach's α coefficient of 0.944 for the entire scale, with coefficients above 0.696 for each dimension and a split-half reliability of 0.891. These indicators confirm the scale's high internal consistency and stability. In summary, The adapted RS-KTPL scale exhibits good reliability and validity, providing a scientific basis for clinicians to assess patients' psychological resilience and formulate interventions. thereby formulating more effective nursing interventions.

In this study, the adapted Chinese version of the RS-KTPL scale demonstrated superior reliability and validity to the original scale. Specifically, the internal consistency reliability (Cronbach's α coefficient of 0.944) was higher than that of the original scale (0.87) [2]. In terms of validity, the content validity indices of the adapted scale (I-CVI ranging from 0.83 to 1.000, S-CVI/Ave of 0.91) were higher than those of the original scale (I-CVI ≥ 0.80, S-CVI/Ave ≥ 0.90) [2].Structural validity analysis confirmed the four-dimensional structure through exploratory and confirmatory factor analyses, with a cumulative variance contribution rate of 64.913%, surpassing the 50.71% of the original scale [2]. The analysis of convergent and discriminant validity showed that the adapted scale had CR values ranging from 0.741 to 0.938, AVE values ranging from 0.5 to 0.704, and the square roots of AVE were greater than the correlation coefficients, demonstrating good convergent and discriminant validity. In contrast, although the original scale met acceptable standards, it performed less robustly in certain dimensions than the adapted scale. This suggests that the adapted scale has a more rational structure and more comprehensively reflects the psychological resilience of patients who underwent kidney transplantation. This improvement

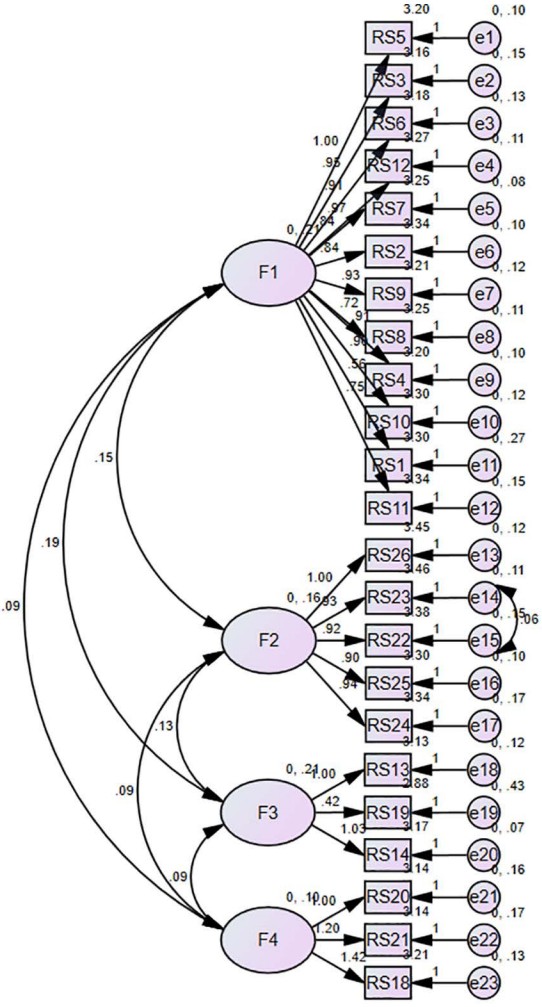

**Fig 2. Confirmatory factor analysis path diagram with standardized loadings.**

**Table 4. Model fit indices for the confirmatory factor analysis.**

| Fit Index | Recommended Criteria | Obtained Value | Interpretation |
|---|---|---|---|
| $\chi^2$/df (CMIN/DF) | <3.0 | 2.054 | Excellent |
| RMSEA | <0.08 | 0.071 | Excellent |
| SRMR | <0.08 | 0.016 | Excellent |
| PGFI | >0.50 | 0.677 | Excellent |
| CFI | >0.90 | 0.920 | Excellent |
| TLI | >0.90 | 0.910 | Excellent |
| IFI | >0.90 | 0.921 | Excellent |

*Note. $\chi^2$/df = Chi-square to degrees of freedom ratio; RMSEA = Root Mean Square Error of Approximation; SRMR = Standardized Root Mean Square Residual; PGFI = Parsimony Goodness-of-Fit Index; CFI = Comparative Fit Index; TLI = Tucker-Lewis Index; IFI = Incremental Fit Index.*

**Table 5. Composite reliability and convergent validity of the Chinese version of the psychological Resilience Scale for Kidney Transplant patients.**

| Dimension | Item | Standardized Loading | Composite Reliability (CR) | AVE |
|---|---|---|---|---|
| **Positive Mindset** | | | 0.938 | 0.559 |
| | RS5 | 0.827 | | |
| | RS3 | 0.720 | | |
| | RS6 | 0.773 | | |
| | RS12 | 0.827 | | |
| | RS7 | 0.797 | | |
| | RS2 | 0.760 | | |
| | RS9 | 0.770 | | |
| | RS8 | 0.680 | | |
| | RS4 | 0.816 | | |
| | RS10 | 0.779 | | |
| | RS1 | 0.478 | | |
| | RS11 | 0.680 | | |
| **Medical Management** | | | 0.851 | 0.534 |
| | RS26 | 0.749 | | |
| | RS23 | 0.753 | | |
| | RS22 | 0.696 | | |
| | RS25 | 0.772 | | |
| | RS24 | 0.678 | | |
| **Social Support** | | | 0.826 | 0.704 |
| | RS13 | 0.788 | | |
| | RS14 | 0.887 | | |
| **Healthy Habits** | | | 0.741 | 0.500 |
| | RS20 | 0.645 | | |
| | RS21 | 0.689 | | |
| | RS18 | 0.761 | | |

Note. All factor loadings were significant at $p < 0.001$. AVE = Average Variance Extracted. CR values exceeded the recommended 0.70 threshold and AVE values met the 0.50 criterion, demonstrating good convergent validity. due to the low load, RS19(0.275) of the social support dimension was deleted.

**Table 6. Discriminant validity of the psychological resilience scale.**

| Variable | 1 | 2 | 3 | 4 |
|---|---|---|---|---|
| 1. Social Support | 0.839 | | | |
| 2. Medical Management | 0.703** | 0.731 | | |
| 3. Positive Mindset | 0.911** | 0.798** | 0.748 | |
| 4. Healthy Habits | 0.619** | 0.714** | 0.639** | 0.707 |
| AVE | 0.704 | 0.534 | 0.559 | 0.500 |

Note. Diagonal elements (bold) represent the square root of AVE. Off-diagonal elements are factor correlation coefficients.

**$p < 0.01$. All AVE values met the Fornell-Larcker criterion (square root of AVE > inter-factor correlations), supporting discriminant validity.

may be attributed to the removal of certain inapplicable items, resulting in a more concise structure. Additionally, cultural adaptation optimizations in the adapted scale clarified the distinctions between dimensions, reducing misunderstandings and inconsistencies in responses caused by cultural differences [18]. In conclusion, the adapted Chinese version of the

RS-KTPL scale exhibits superior reliability and validity compared to the original scale and, through cultural adaptation and item refinement, better aligns with the actual conditions of Chinese kidney transplant patients. This provides the clinical field with a more accurate and effective assessment tools.

### Process and rationale for item reduction

The removal of Items 15–17 reflects cultural nuances in psychological reporting, while Item 19's exclusion highlights differences in clinical management between Chinese and Korean contexts.

Patients from different cultural backgrounds may exhibit variations in their psychological resilience and coping mechanisms [19]. Specifically, Items 15–16 (transplant-related anxiety) were removed because Chinese patients' concerns are mitigated by high transplant success rates (92% 5-year survival [20,21] and preoperative counseling, with anxiety more often expressed somatically [22,23]. Item 17 was excluded because Chinese patients tend to attribute stress to familial/social expectations rather than personal recovery, making this item culturally insensitive [23].

Clinically, Item 19 (fluid control) was deemed irrelevant because standardized hydration protocols in Chinese hospitals reduce patient self-regulation needs, contrasting with the original scale's emphasis on dietary autonomy [24].

### Conclusion

The Chinese version of the RS-KTPL scale, adapted for this study, consists of 22 items across four dimensions and demonstrates good reliability and validity. This provides a valuable assessment tool for nursing management personnel to explore effective interventions for enhancing the psychological resilience of kidney transplant patients, ultimately safeguarding their mental health and rehabilitation outcomes. The limitations of this study include the fact that the sample was primarily drawn from a single tertiary hospital, potentially introducing regional and selection bias. Although the scale demonstrated good reliability and validity in this study, its applicability across different racial, cultural, and linguistic backgrounds requires further verification. Future research should conduct multicenter studies to further validate the scale's applicability and perform prospective studies to assess its long-term stability and predictive validity. Further exploration is needed to evaluate the scale's applicability across different cultural contexts to provide more precise clinical guidance for nursing practice.

### Supporting information

**S1 Data. Minimal dataset including participant responses and raw scores.**
(XLSX)

**S1 File. Final Chinese version of the 22-item RS-KTPL scale.**
(DOCX)

### Acknowledgments

The author would like to thank the experts at Wuhan University, Renmin Hospital of Wuhan University, and the Central Hospital of Enshi Tujia and Miao Autonomous Prefecture. and all the participants who graciously devoted their time to participate in the study.

### Author contributions

**Conceptualization:** YAN TANG, Jiefu Zhu, Tao Qiu.

**Data curation:** YAN TANG, Xiaolan Mao.

**Formal analysis:** Xiaolan Mao.

**Funding acquisition:** Jiefu Zhu.

**Investigation:** YAN TANG.

**Methodology:** YAN TANG, Shan Qiu, Wei Lei, Xiaolan Mao.

**Supervision:** Jiefu Zhu, Tao Qiu.

**Validation:** YAN TANG, Xiaolan Mao.

**Writing – original draft:** YAN TANG.

**Writing – review & editing:** YAN TANG, Jiefu Zhu, Tao Qiu, Shan Qiu, Wei Lei, Xiaolan Mao.

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
