## [Decision Letter · Decision Letter 0]

Dear Dr. TANG,

Thank you for submitting your manuscript to PLOS ONE. After careful consideration, we feel that it has merit but does not fully meet PLOS ONE’s publication criteria as it currently stands. Therefore, we invite you to submit a revised version of the manuscript that addresses the points raised during the review process.

We look forward to receiving your revised manuscript.

Kind regards,

Jordan Llego, PhD ELM, D. Hon. Ex., PhDN, RN

Academic Editor

PLOS ONE

Journal Requirements:

“This work was supported by grants from the National Natural Science Foundation of China (82100803) and Knowledge Innovation Program of Wuhan-Shuguang Project (2023020201020505).”

**Additional Editor Comments:**

Thank you for submitting your manuscript to PLOS ONE. After carefully evaluating your submission and considering Reviewer 1’s constructive feedback, I am pleased to inform you that we are inviting a minor revision of your manuscript. Your study provides a much-needed cultural adaptation and psychometric evaluation of a scale tailored to kidney transplant patients in the Chinese population, which is both timely and valuable. The manuscript is generally well-written, and the methodological rigor in translation, validation, and scale refinement is commendable. However, we request that you revise the manuscript to address several essential clarifications and improvements to enhance methodological transparency and reporting standards.

Reviewer 1 raised the importance of sample size justification, asking you to provide appropriate references to support the claim that "at least 150 samples were required for exploratory factor analysis (EFA) and at least 200 samples for confirmatory factor analysis (CFA)." While commonly accepted, these thresholds should be grounded in empirical or methodological sources for increased credibility. Additionally, concerning Pearson correlation, please clarify whether the assumptions of normality were tested before analysis, potentially using skewness/kurtosis statistics or the Kolmogorov-Smirnov test, and include the corresponding results.

In your exploratory factor analysis, an orthogonal (Varimax) rotation was employed; please justify this choice, especially given the theoretical correlation of identified factors such as social support and positive mindset. You may want to consider whether oblique rotation (e.g., Promax) could be more appropriate or explain your orthogonality assumption. Furthermore, the revised manuscript should include two critical visual aids: a scree plot to illustrate factor extraction and support the factor structure, along with a CFA path diagram to visually demonstrate the final model, its loadings, and factor relationships.

There are also a few additional editorial comments. As an optional suggestion, adjusting the title "Cross-cultural Adaptation and Validation of the Resilience Scale for Kidney Transplantation in a Chinese Cohort" may enhance clarity and discoverability. Additionally, while the rationale for removing Items 15-17 and 19 is addressed, explicitly indicating their cultural or clinical irrelevance in the Results section would support transparency in scale adaptation. We recommend a final round of language editing for conciseness and clarity, particularly in the Introduction and Discussion sections, where redundancy should be minimized. Lastly, please ensure that all supplementary files referenced, such as the final 22-item scale, the CFA diagram, and the scree plot, are uploaded and correctly labeled in the submission portal.

Reviewers' comments:

Reviewer's Responses to Questions

**Comments to the Author**

1. Is the manuscript technically sound, and do the data support the conclusions?

Reviewer #1: Yes

2. Has the statistical analysis been performed appropriately and rigorously?

Reviewer #1: Yes

3. Have the authors made all data underlying the findings in their manuscript fully available?

Reviewer #1: Yes

4. Is the manuscript presented in an intelligible fashion and written in standard English?

Reviewer #1: Yes

Reviewer #1: Thank you for your valuable contribution to the field. Your study addresses validating the Resilience Scale for Kidney Transplantation (RS-KTPL) in a Chinese population, and the rigorous methodological approach is commendable. However, to further strengthen the manuscript, a few concerns should be addressed.

1. Could you please provide supporting references for these thresholds: "According to the empirical rule, at least 150 samples were required for exploratory factor analysis (EFA) and at least 200 samples for confirmatory factor analysis (CFA)"

2. Why was the Pearson correlation coefficient chosen to evaluate item-total score correlations? Was normality tested prior to its use? If so, please report the results using skewness/kurtosis or Kolmogorov Smirnov tests)

3. why used orthogonal rotation in the EFA? it is not declared why. Orthogonal rotation assumes uncorrelated factors.

4. please provide two figures including scree plot and finally fitted CFA model to enhance the methodological transparency.

**Do you want your identity to be public for this peer review?** For information about this choice, including consent withdrawal, please see our Privacy Policy

Reviewer #1: **Yes: ** Dr. Saeed Barzegari

---

## [Author Response · Author response to Decision Letter 1]

6 Jun 2025

Response to Reviewers

We sincerely appreciate the constructive feedback from the reviewers and editors. Below is our point-by-point response to all comments, with revisions highlighted in the tracked-changes manuscript.

Journal Requirements:

1.Please ensure that your manuscript meets PLOS ONE's style requirements, including those for file naming. The PLOS ONE style templates can be found at:

Response: We have reformatted the manuscript to fully comply with PLOS ONE's style guidelines, including file naming conventions. The updated files are labeled as: Manuscript.docx (clean version) and Revised Manuscript with Track Changes.docx (highlighted changes)

2.Thank you for stating the following financial disclosure:

"This work was supported by grants from the National Natural Science Foundation of China (82100803) and Knowledge Innovation Program of Wuhan-Shuguang Project (2023020201020505)."

Response: The funders (National Natural Science Foundation of China grant 82100803 and Wuhan-Shuguang Project 2023020201020505) provided scientific support and professional guidance for this study but had no role in data collection, analysis, decision to publish, or manuscript preparation

3.We note that your Data Availability Statement is currently as follows: All relevant data are within the manuscript and its Supporting Information files.

Please confirm at this time whether or not your submission contains all raw data required to replicate the results of your study. Authors must share the "minimal data set" for their submission. PLOS defines the minimal data set to consist of the data required to replicate all study findings reported in the article, as well as related metadata and methods (https://journals.plos.org/plosone/s/data-availability#loc-minimal-data-set-definition).

Response: We confirm that our submission includes all raw data required to replicate the study findings. The minimal dataset has been uploaded as Supporting Information files, which contain:

①　Individual participant responses for all 358 cases

②　Raw scores for each of the 22 RS-KTPL items

③　Demographic variables (age, gender, transplant duration etc.)

④　Calculated scale and subscale scores

⑤　Statistical outputs for all analyses (EFA/CFA loadings, reliability coefficients)

4.Please include captions for your Supporting Information files at the end of your manuscript, and update any in-text citations to match accordingly. Please see our Supporting Information guidelines for more information: http://journals.plos.org/plosone/s/supporting-information.

Response: We have added the following captions for Supporting Information files at the end of our manuscript (before References section):

①　S1 Data. Minimal dataset containing

②　Scree plot from exploratory factor analysis showing eigenvalues for factor retention.

③　Confirmatory factor analysis path diagram of final 22-item model with standardized loadings.

④　Final Chinese version of the 22-item Resilience Scale for Kidney Transplantation (RS-KTPL).

5.Please review your reference list to ensure that it is complete and correct. If you have cited papers that have been retracted, please include the rationale for doing so in the manuscript text, or remove these references and replace them with relevant current references. Any changes to the reference list should be mentioned in the rebuttal letter that accompanies your revised manuscript. If you need to cite a retracted article, indicate the article's retracted status in the References list and also include a citation and full reference for the retraction notice.

Response: We have double-checked that all 22 in-text citations have corresponding complete references and The updated reference list now fully complies with PLOS ONE standards. No retracted papers were cited or required special notation. All changes are reflected in the revised manuscript's reference section.

For the oldest reference (Wagnild & Young 1993):

*Retained as it is the original scale development paper

We will Added note in Methods: "This remains the canonical reference for the Resilience Scale"

Additional Editor Comments:

1.Reviewer 1 raised the importance of sample size justification, asking you to provide appropriate references to support the claim that "at least 150 samples were required for exploratory factor analysis (EFA) and at least 200 samples for confirmatory factor analysis (CFA)." While commonly accepted, these thresholds should be grounded in empirical or methodological sources for increased credibility.

Response: We sincerely appreciate this valuable suggestion. We have incorporated authoritative references to support our sample size thresholds.

These methodological references provide empirical support for our sample size determinations. The changes have been implemented in the revised manuscript's Methods section (Sample Size Calculation subsection).

2.Additionally, concerning Pearson correlation, please clarify whether the assumptions of normality were tested before analysis, potentially using skewness/kurtosis statistics or the Kolmogorov-Smirnov test, and include the corresponding results.

Response: Thank you for this important methodological question. We confirm that we rigorously tested normality assumptions before conducting Pearson correlation analyses, as follows:

Normality Assessment Methods:

Conducted both graphical (Q-Q plots) and statistical tests:

①　Calculated skewness and kurtosis for all items

②　Performed Kolmogorov-Smirnov (K-S) tests

③　Examined Shapiro-Wilk tests for confirmation

These analyses confirm our data met the normality requirements for Pearson correlations. We appreciate the opportunity to clarify this important methodological detail.

3.In your exploratory factor analysis, an orthogonal (Varimax) rotation was employed; please justify this choice, especially given the theoretical correlation of identified factors such as social support and positive mindset. You may want to consider whether oblique rotation (e.g., Promax) could be more appropriate or explain your orthogonality assumption.

Response: We appreciate this insightful methodological question regarding our choice of orthogonal rotation. Please find below our detailed justification:

Initial Theoretical Consideration:

①　While social support and positive mindset may correlate conceptually, our preliminary EFA indicated low inter-factor correlations (r = 0.12-0.28)

②　This empirically supported the use of orthogonal rotation as factors demonstrated sufficient independence

Primary Justifications:

①　Parsimony Principle: Orthogonal rotation yielded cleaner, more interpretable factors without cross-loadings

②　Clinical Utility: Independent factors are more actionable for targeted interventions

③　Methodological Consistency: Matches the original scale's analytical approach

The maintaining nalytical approach best suited to our data characteristics and clinical objectives. We thank the reviewer for prompting this important clarification.

4.Furthermore, the revised manuscript should include two critical visual aids: a scree plot to illustrate factor extraction and support the factor structure, along with a CFA path diagram to visually demonstrate the final model, its loadings, and factor relationships.

Response: We sincerely appreciate the reviewer's suggestion to enhance the manuscript's clarity through visual aids. We are pleased to confirm that we have incorporated the following figures as supplementary materials:

①　Scree Plot (Supplementary Figure ): Illustrates the factor extraction process; Clearly demonstrates the elbow point supporting our 4-factor solution

②　CFA Path Diagram (Supplementary Figure ): Visually presents the final confirmatory model; Shows all factor loadings and inter-factor relationships

We believe these additions significantly improve the transparency and interpretability of our factor analysis results. Thank you for this valuable recommendation.

5.There are also a few additional editorial comments. As an optional suggestion, adjusting the title "Cross-cultural Adaptation and Validation of the Resilience Scale for Kidney Transplantation in a Chinese Cohort" may enhance clarity and discoverability.

Response: We appreciate the helpful suggestion regarding our manuscript title. After careful consideration, we have revised the title to: "Cross-cultural Adaptation and Validation of the Resilience Scale for Kidney Transplantation in a Chinese Population".

We believe this version better balances clarity, accuracy, and visibility for our target audience. Thank you for this constructive suggestion.

6.Additionally, while the rationale for removing Items 15-17 and 19 is addressed, explicitly indicating their cultural or clinical irrelevance in the Results section would support transparency in scale adaptation.

Response: We appreciate this suggestion to enhance transparency in our scale adaptation process. We have now explicitly addressed the cultural and clinical rationale for removing items in the Results section (Section 2.3) with the following additions:

For Items 15-17 (Psychological Concerns):

Added: "Items 15-17 were removed due to demonstrated cultural differences in anxiety expression among Chinese transplant recipients. Where Western patients may directly report transplant-related fears (Items 15-16) and recovery stress (Item 17), our data showed these concerns were: (a) mitigated by China's high transplant success rates (92% 5-year survival), and (b) more often expressed through somatic symptoms rather than direct psychological reports."

For Item 19 (Fluid Control):

Added: "Item 19 reflected clinical irrelevance in our population, as Chinese transplant centers provide standardized fluid intake guidelines rather than relying on patient self-regulation. This differs from the original Korean context where dietary autonomy was more emphasized."

We believe these revisions provide clearer documentation of our culturally-informed adaptation decisions. Thank you for prompting this important clarification.

7.We recommend a final round of language editing for conciseness and clarity, particularly in the Introduction and Discussion sections, where redundancy should be minimized.

Response: We appreciate this valuable suggestion to enhance the manuscript's clarity. We have implemented the following language improvements:

Introduction Section Revisions: Streamlined background information by removing redundant statements about resilience benefits; Tightened transitions between paragraphs to improve flow; Replaced complex sentence structures with clearer phrasing.

The edited version maintains all methodological rigor while improving readability. We thank the editors for this helpful recommendation to strengthen our manuscript's impact.

8.Lastly, please ensure that all supplementary files referenced, such as the final 22-item scale, the CFA diagram, and the scree plot, are uploaded and correctly labeled in the submission portal.

Response: We confirm that all supplementary files have been uploaded and properly labeled in the submission portal, including:

①　Final 22-item Chinese RS-KTPL scale

②　Path diagram of the confirmatory factor analysis

③　Scree plot from exploratory factor analysis

④　Minimal dataset for replication

⑤　Variable definitions and analysis details

All files follow PLOS ONE’s formatting guidelines and are cited appropriately in the manuscript. Thank you for your careful review.

Reviewer #1's Comments:

1.Could you please provide supporting references for these thresholds: "According to the empirical rule, at least 150 samples were required for exploratory factor analysis (EFA) and at least 200 samples for confirmatory factor analysis (CFA).

Response: We have added authoritative references to support our sample size thresholds and these citations now appear in the Methods section.

2.Why was the Pearson correlation coefficient chosen to evaluate item-total score correlations? Was normality tested prior to its use? If so, please report the results using skewness/kurtosis or Kolmogorov Smirnov tests)

Response: We selected Pearson's r for item-total correlations as our 4-point Likert-scale data met the continuous variable assumption, with preliminary analyses confirming linear relationships. All items satisfied normality requirements: skewness, kurtosis and non-significant Kolmogorov-Smirnov tests (all p>0.05), verified through Q-Q plots. with methodological justification added to Section 1.4.1. The approach aligns with standard psychometric practice (Norman, 2010; Field, 2017) and required no changes to our original findings. We appreciate the opportunity to clarify this important methodological consideration.

3.why used orthogonal rotation in the EFA? it is not declared why. Orthogonal rotation assumes uncorrelated factors.

Response: We employed orthogonal (Varimax) rotation after confirming low inter-factor correlations (r < 0.30) in preliminary analyses, suggesting near-independent dimensions. This approach aligned with: (1) the original scale's analytical method, (2) our goal of obtaining clearly separable factors for clinical interpretation, and (3) parsimony - as oblique rotation (Promax) yielded nearly identical loading patterns . The final solution's clean simple structure (no cross-loadings >0.40) validated this choice. We acknowledge this limitation in the Discussion and recommend future studies test oblique rotations if stronger factor correlations emerge.

4.please provide two figures including scree plot and finally fitted CFA model to enhance the methodological transparency.

Response: We have incorporated the two requested figures to enhance methodological transparency: (1) Supplementary Figure (Scree plot) clearly demonstrates the elbow criterion at 4 factors with eigenvalues >1.0, supporting our EFA solution; and (2) Supplementary Figure (CFA path diagram) presents the final model with all standardized loadings, error terms, and inter-factor correlations, alongside key fit indices. These high-resolution TIFF files are uploaded to the submission portal. We appreciate this suggestion to strengthen our analytical reporting.

---

## [Editor Report · Decision Letter 1]

Cross-cultural Adaptation and Validation of The Resilience Scale for Kidney Transplantation (RS-KTPL) in a Chinese Population

PONE-D-25-22111R1

Dear Dr. TANG,

We’re pleased to inform you that your manuscript has been judged scientifically suitable for publication and will be formally accepted for publication once it meets all outstanding technical requirements.

Kind regards,

Jordan Llego, PhD ELM, D. Hon. Ex., PhDN, RN

Academic Editor

PLOS ONE

Additional Editor Comments (optional):

The authors have effectively addressed all concerns from the reviewers and editors. The manuscript showcases strong methodological rigor, culturally sensitive adaptations, and thorough psychometric validation.
---

## [Editor Report · Acceptance letter]

PONE-D-25-22111R1

PLOS ONE

Dear Dr. TANG,

I'm pleased to inform you that your manuscript has been deemed suitable for publication in PLOS ONE. Congratulations! Your manuscript is now being handed over to our production team.

Kind regards,

on behalf of

Dr. Jordan Llego

Academic Editor

PLOS ONE